# Genome-Wide Identification and Expression Analyses of the Aquaporin Gene Family in Passion Fruit (*Passiflora edulis*), Revealing *PeTIP3-2* to Be Involved in Drought Stress

**DOI:** 10.3390/ijms23105720

**Published:** 2022-05-20

**Authors:** Shun Song, Dahui Zhang, Funing Ma, Wenting Xing, Dongmei Huang, Bin Wu, Jian Chen, Di Chen, Binqiang Xu, Yi Xu

**Affiliations:** 1Key Laboratory of Genetic Improvement of Bananas, Haikou Experimental Station/Hainan Key Laboratory for Biosafety Monitoring and Molecular Breeding in Off-Season Reproduction Regions, Sanya Research Institute of Chinese Academy of Tropical Agricultural Sciences, Chinese Academy of Tropical Agricultural Sciences, Haikou 571101, China; sss1984006@163.com (S.S.); funm123@126.com (F.M.); wtx@163.com (W.X.); dongmeih@163.com (D.H.); wubin885@163.com (B.W.); cd76@126.com (D.C.); binqiangx@163.com (B.X.); 2Hainan Yazhou Bay Seed Laboratory, Sanya 571101, China; 3Yunnan Agricultural University, Kunming 650201, China; dahuizhang@126.com (D.Z.); chenjian112@126.com (J.C.)

**Keywords:** aquaporin (AQP), abiotic stress, passion fruit, expression analysis, fruit maturity stage

## Abstract

Aquaporins (AQPs) in plants can transport water and small molecules, and they play an important role in plant development and abiotic stress response. However, to date, a comprehensive study on AQP family members is lacking. In this study, 27 AQP genes were identified from the passion fruit genome and classified into four groups (NIP, PIP, TIP, SIP) on the basis of their phylogenetic relationships. The prediction of protein interactions indicated that the AQPs of passion fruit were mainly associated with AQP family members and boron protein family genes. Promoter cis-acting elements showed that most PeAQPs contain light response elements, hormone response elements, and abiotic stress response elements. According to collinear analysis, passion fruit is more closely related to Arabidopsis than rice. Furthermore, three different fruit ripening stages and different tissues were analyzed on the basis of the transcriptome sequencing results for passion fruit AQPs under drought, high-salt, cold and high-temperature stress, and the results were confirmed by qRT-PCR. The results showed that the PeAQPs were able to respond to different abiotic stresses, and some members could be induced by and expressed in response to multiple abiotic stresses at the same time. Among the three different fruit ripening stages, 15 AQPs had the highest expression levels in the first stage. AQPs are expressed in all tissues of the passion fruit. One of the passion fruit aquaporin genes, *PeTIP3-2*, which was induced by drought stress, was selected and transformed into Arabidopsis. The survival rate of transgenic plants under drought stress treatment is higher than that of wild-type plants. The results indicated that *PeTIP3-2* was able to improve the drought resistance of plants. Our discovery lays the foundation for the functional study of AQPs in passion fruit.

## 1. Introduction

Passiflora (Passiflora edulis) is a perennial evergreen vine of the passiflora genus (passifloraceae), a tropical and rare fruit tree. Passion fruit is named after its fruit pulp, which contains the aroma of over 100 different fruits. It is native to central and northern South America, and is widely distributed throughout Central and South America, Australia, and Africa [1]. According to reports, there are about 520 species of passiflora, most of which are cultivated for flower viewing and gardening, and only about 60 species can be eaten fresh [2]. At present, the main countries in which passion fruit is grown are Brazil, Colombia, Ecuador, Australia, Vietnam, China, etc. The planting area is gradually expanding due to its distinct flavor and short growth cycle of 4–6 months. Drought, high salinity, high temperature, cold, and other abiotic stresses seriously affect the normal growth and development of passion fruit in the main producing areas, resulting in a significant decrease in yield and fruit quality. Therefore, mining the stress-resistance-related functional genes of passion fruit, identifying their functions, and analyzing their mechanisms of action are very important for improving the varieties.

A large number of studies have reported that membrane proteins in plant cells play an important role in the process of plant stress resistance. Aquaporins (AQPs) are one of the most deeply and systematically studied types of membrane proteins. They belong to the major intrinsic protein (MIP) superfamily, which is widely present in humans, animals, plants, fungi, and bacteria, and which play a role in the transport of water and other small molecules such as glycerol, CO_2_, and boron through biological membranes [3,4]. In most plant species, AQPs can be divided into eight categories on the basis of gene sequence homology and subcellular location, including tonoplast intrinsic proteins (TIPs), plasma membrane intrinsic proteins (PIPs), small basic intrinsic proteins (SIPs), nodulin 26-like intrinsic proteins (NIPs), hybrid intrinsic proteins (HIPs), GlpF-like intrinsic proteins (GIPs), large intrinsic proteins (LIPs), and uncategorized members, which are designated as X intrinsic proteins (XIPs) [5,6]. Plant AQPs possess conserved structural characteristics, including Asn-Pro-Ala (NPA) motifs, an aromatic/arginine (ar/R) region, and a trans-membrane domain in which six membrane-spanningα-helices are linked by five short loops with their N- and C-terminus towards the cytosol [6,7].

Related research reports have shown that AQPs effectively mediate the rapid transmembrane transport of water and play a critical role during plant growth and development processes [8], such as the fruit ripening process. According to related research, the strawberry aquaporin *FaPIP2;1* is involved in the fruit ripening process [9]. The aquaporin genes *VvPIP1a* and *VvPIP1b* in grapes are highly expressed at various stages of fruit development, indicating that they are involved in the process of fruit development [10]. A total of 47 aquaporin genes have been cloned in each developmental stage of tomato [11].

In recent years, the function of AQPs has been further related to the resistance of plants [12,13]. It has been demonstrated that AQP can improve the tolerance of plants to abiotic stresses (such as drought, osmosis, cold, salt, and high temperature) [4,6,7,14,15,16,17]. Tobacco *NtAQP1* can improve plant water absorption capacity under high salt stress [18]. Overexpression of *SpAQP1*, *TdPIP1;1*, *OsPIP1;1*, *SlTIP2;2,* and *TaNIP* resulted in a salt-tolerant phenotype in transgenic plants [19,20]. *TaTIP2;2* in wheat increases drought stress tolerance in transgenic Arabidopsis [21]. Similarly, *MaPIP1;1* enhances drought and salt tolerance in transgenic Arabidopsis [22]. *MaPIP2-7* and *MaSIP2-1* improves the adaptability of transgenic bananas to drought, salt, and cold [4,6]. The expression of *FaPIPs*, *SvPIPs*, *RsPIPs,* and *RsTIPs* can be induced under heat stress [23,24,25]. In Arabidopsis, *AtPIPs* were highly up-regulated due to combined heat and drought stress [26]. These studies indicate that AQP plays a positive role under abiotic stress [13].

More than 1200 AQP genes have been identified in 31 plant species, including 35 AQP members in Arabidopsis [27], 47 in tomatoes [11], 36 in maize [28], 33 in rice [29], 47 in bananas [30], and 53 in Chinese cabbage [31]. Whole-genome sequencing of passion fruit enables us to perform genome-wide analysis of its genes [32]. However, the function of PeAQPs in passion fruit remains poorly understood.

In this study, we identified and analyzed the PeAQPs with respect to their phylogenetic relationship, gene structure, protein motifs, protein features, sequence phylogeny, chromosomal location, promoter elements, and so on. More importantly, we also identified the expression patterns of the genes during the fruit maturity stage and the abiotic stress, which were also validated by qPCR analyses. One of the aquaporin genes whose expression was highly induced by drought stress was chosen for further functional verification. The results showed that it was able to improve the drought tolerance of plants. This provides useful information on the genetic improvement of passion fruit quality and resistance to abiotic stress, and lays a good foundation for the regulating mechanisms of PeAQPs.

## 2. Results

### 2.1. Identification of the Passion Fruit AQP Family

On the basis of HMM, protein BLAST searches, and the use of the MEME software, a total of 27 PeAQP members were identified and annotated from the passion fruit genome database. In addition, we analyzed the characteristics of the PeAQPs (Table 1); the length of the PeAQP CDS sequence ranged from 234 bp for *PeNIP6-1* to 1794 bp for *PeSIP2-1*. The identified PeAQPs encoded proteins ranging from 78 amino acids in the case of *PeNIP6-1* to 598 amino acids in the case of *PeSIP2-1*. With respect to the phosphorylation site analyses, 89% of PeAQPs were found to contain all three phosphorylation sites (Ser, Thr and Tyr). Among them, *PeNIP6-1*, *PeSIP2-1,* and *PeSIP2-2* only contained the Ser and Thr phosphorylation sites. With respect to subcellular location prediction analysis, except for *PeNIP6-1*, located in the nucleus, all other AQP members were located on the cell membrane.

### 2.2. Evolutionary Characterization of AQP Genes

To study the classification and evolutionary relationship of AQP proteins in passion fruit, a phylogenetic tree was constructed with full-length protein sequences of the aquaporin gene family members in Arabidopsis, rice, and passion fruit (Figure 1).

The PeAQPs in passion fruit can be divided into four groups (PIP, NIP, TIP, and SIP). There are nine PIPs, seven NIPs, eight TIPs, and three SIPs. Among the PIP subfamily, nine members were divided into two groups: PIP1 with two members and PIP2 with seven members. Four groups were found for the TIP subfamily (TIP1 to TIP4), with four members in the TIP1 group, two members in each of the TIP2 and TIP3 groups, and one member in the TIP4 group. Four groups belonged to the NIP subfamily, with two members in the NIP1 group, three members in the NIP4 group, and one member in each of the NIP5 and NIP6 groups.

Passion fruit and Arabidopsis are both dicotyledonous plants. The hypothetical homologous members of AQPs in passion fruit and the known AQPs of Arabidopsis can be inferred. Among PIP subfamily genes, *PePIP2-11* was the best orthologous match of *AtPIP2-7* and *AtPIP2-8*; *PePIP2-8* exhibited the closest relationship with *AtPIP2-1*, *AtPIP2-2*, *AtPIP2-3*, *AtPIP2-4,* and *AtPIP2-6*; *PePIP1-5* was the most homogeneous gene of *AtPIP1-1* and *AtPIP1-2*. For NIP subfamily genes, *PeNIP1-1* and *PeNIP1-2* were phylogenetically closest to *AtNIP1-1* and *AtNIP1-2*; *PeNIP6-1* exhibited the closest relationship with *AtNIP6-1*; and *PeNIP5-1* was closely related to *AtNIP5-1*. For TIP subfamily genes, *PeTIP3-1* and *PeTIP3-2* were the most homogeneous genes of *AtTIP3-2**; PeTIP2-1* was closely related to *AtTIP2-1*; and *PeTIP4-1* exhibited the closest relationship with *AtTIP4-1*. For SIP subfamily genes, *PeSIP2-1* and *PeSIP2-2* were closely related to *AtSIP2-1**;* and *PeSIP1-1* was the most homogeneous gene of *AtSIP1-1* and *AtSIP1-2*.

Because rice and passion fruit are more distantly related, as shown in the figure, they have fewer close pairings. For PIP subfamily genes, only one member, *PePIP1-2*, was the best orthologous match of *OsPIP1-2* and *OsPIP1-3*. For NIP subfamily genes, *PeNIP4-3* was closely related to *OsNIP2-1* and *OsNIP2-2. PeNIP5-1* was the most homogeneous gene of *OsNIP3-1*. For TIP subfamily genes, *PeTIP1-1* was phylogenetically closest to *OsTIP1-1*.

### 2.3. Gene Structure and Conserved Motif Analysis

A total of 15 conserved motifs in PeAQPs were predicted using MEME software (Figure 2). With respect to the homology of the genes, their recognition motifs were also similar. In the PIPs group, most members had nine similar motifs, except for *PePIP2-7*, which had seven motifs, and *PePIP1-2* and *PePIP1-5*, which had eight motifs. In the TIPs group, all members had motifs 1, 2, 6, 7, 8, 9, and 10 except *PeTIP4-1*, which had motifs 1, 2, 6, 7, 8, 9, and 11. In the NIPs group, *PeNIP4-1* and *PeNIP4-2*, and *PeNIP1-1,* and *PeNIP1-2* had the same motifs. Only *PeNIP6-1* had two motifs. In the SIPs group, *PeSIP2-1* and *PeSIP2-2* had the same motifs, including 1, 3, 4, 8, 10, and 11. *PeSIP1-1* had the motifs 1, 3, and 8.

The exon–intron structure is an important evolutionary feature of genes. We also further analyzed the exon–intron organization of PeAQP family members. Most of the members had structures of 5′UTR (untranslated region) and 3′UTR, except that of *PeTIP1-3* and *PeNIP6-1,* which had no UTR structure. In addition, *PePIP2-2* and *PeTIP2-2* had only the 5′UTR structure. *PeSIP1-1* had only 3′UTR. Most family members contained 2–4 exons, whereas *PeNIP6-1* had only one exon (Figure 2). The diverse gene structures in different subgroups of passion fruit resulted in the genes evolving into diverse exon–intron structures to perform different functions.

### 2.4. Chromosome Distribution of AQP Genes

The genomic distribution of each passionfruit AQP was investigated, as indicated in Figure 3. The 27 PeAQPs were located on seven chromosomes, which were chromosomes 1, 2, 3, 5, 6, 8, and 9. Among them, Chr1, Chr3, and Chr9 presented seven, two, and five PeAQPs, respectively. Chr2, Chr5, Chr6, and Chr8 each presented three PeAQPs, whereas one PeAQP (*PePIP1-2*) was located at unknown chromosomal positions. The PIPs were located on chromosomes 1, 2, 6, and 8. NIPs were positioned on chromosomes 1, 2, 3, 5, 8, and 9. TIPs were located on chromosomes 1, 5, 6, and 9. In addition, SIPs were positioned on chromosomes 3 and 9.

### 2.5. Promoter Analysis of PeAQP Genes

Predicting the cis-regulatory elements in PeAQP gene promoters can help to better understand the transcriptional regulatory functions of gene family members (Figure 4). Among them, the TATA-box and the CAAT-box could be found in 27 PeAQPs. Moreover, the promoter region of this family gene contained a large number of action elements related to abiotic stress, including drought-responsive elements (CAACTG), cold-responsive elements (CCGAAA), and salicylic acid-responsive elements (TCAGAAGAGG and CCATCTTTTT). Additionally, there were some hormone-related elements. Abscisic acid-responsive elements (ABREs) such as GACACGTGGC and ACGTG, MeJA-responsive motifs such as CGTCA and TGACG, gibberellin-responsive motifs (TATCCCA, CCTTTTG and TCTGTTG), and auxin-responsive elements (AACGAC) were present in abundance. This indicated that the PeAQP gene was regulated by stress response and was involved in plant growth and development.

### 2.6. Interaction Network Analysis of PeAQPs

To better understand the biological functions and regulatory networks involved in PeAQPs, the orthology-based method was used to predict the protein interaction relationships between them. The results showed that 13 PeAQPs had an orthologous relationship with Arabidopsis thaliana, and 13 interacting proteins were found. Most of the proteins that interacted with PeAQPs were aquaporin family genes, whereas the others included boron protein family genes (*BOR1*, *BOR4*) (Figure 5); this is consistent with the function of aquaporins in transporting boron ions. These protein–protein interaction networks provide clues for studying the function of PeAQPs and a basis for further research of candidate genes.

### 2.7. Homology Modeling

The 3D structures of these PeAQP proteins were further predicted using homology-based modeling (Figure 6). The structures of most of the PeAQPs were similar, and contained α-helices (the blue part), an extension chain (the red part), and an irregular curl (the orange part), except that of *PeNIP4-1*, *PeNIP4-3*, *PePIP1-5*, *PeTIP3-1*, *PeTIP3-2*, *PeTIP4-1*, *PeSIP2-1*, *PeSIP2-2,* and *PePIP1-2*, which only contained α-helices (the blue part) and the irregular curl (the orange part); *PeSIP1-1* only contained α-helices (the blue part) and the extension chain (the red part); *PeTIP1-1* and *PeTIP2-1* only contained α-helices (the blue part); *PeNIP6-1* only contained the irregular curl (the orange part); and *PeTIP2-2* contained α-helices (the blue part).

### 2.8. Collinearity Analysis

The collinearity diagrams on the PeAQPs were further analyzed. All genes were displayed on the chromosome circle except *PePIP1-2*. Nine collinearity pairs were found. Most PeAQP genes were found to have one paralogous gene, except for *PePIP2-12*, which had two homologous genes in passion fruit (Figure 7).

In addition, we also performed collinearity analysis on AQP members in passion fruit, Arabidopsis, and rice (Figure 8). There were 12,406 and 3118 genes in Arabidopsis and rice, respectively, that had a collinearity relationship with the genes in passion fruit. Among the 27 members of the PeAQPs, each of them had a collinear relationship with members in Arabidopsis. This is consistent with the evolutionary relationship between passion fruit and Arabidopsis, which are both dicotyledonous plants. Among them, seven genes (*PePIP2-12*, *PePIP2-2*, *PeTIP1-2*, *PeTIP1-1*, *PeTIP4-1*, *PeNIP1-1*, and *PeNIP1-2*) had two homologous genes in Arabidopsis, whereas five genes (*PePIP2-11*, *PePIP2-8*, *PeTIP2-1*, *PeTIP2-2*, and *PeSIP1-1*) had three orthologous genes in Arabidopsis. In addition, three genes (*PePIP2-8*, *PeNIP1-1,* and *PeNIP4-3*) had one orthologous gene in rice. These results help us to further understand the evolutionary relationship of AQPs in passion fruit.

### 2.9. Expression Pattern of AQPs in Passion Fruit

The expression profiles of PeAQPs under various abiotic stresses were investigated using RNA-seq data (Figure 9). The results showed that the four subfamily genes of AQP have different degrees of response to various abiotic stresses. Most genes are induced by drought stress, such as *PeNIP1-2*, *PeNIP6-1*, *PePIP1-2*, *PePIP1-5*, *PeSIP1-1*, *PeSIP2-2*, *PeTIP1-2*, *PeTIP2-2*, *PeTIP3-1*, and *PeTIP3-2*. The transcript levels of *PePIP2-9* and *PePIP2-10* reached their highest levels when the soil moisture was 10%. Some of the genes were induced with increasing degree of salt stress. Among them, *PePIP2-2*, *PePIP2-11*, *PePIP2-8*, *PeTIP1-1*, *PeTIP2-1*, *PeNIP4-3,* and *PeSIP2-1* showed a downward trend. The transcript levels of *PePIP2-7*, *PeTIP1-3,* and *PeNIP4-1* did not change much with increasing salt stress. Under cold stress, only some genes were upregulated, such as *PePIP1-5*, *PeTIP3-1*, *PeTIP3-2*, *PeTIP2-2*, *PeNIP1-1*, and *PeSIP2-2*. In addition, the expressions of *PeTIP1-3*, *PeNIP4-1*, *PeNIP4-2*, *PeNIP5-1*, *PeSIP1-1,* and *PeNIP6-1* exhibited no significant changes. Under high-temperature stress, the transcription levels of most genes were upregulated. Among them, the expression of some genes was induced with increasing stress, such as *PeTIP3-1*, *PeTIP3-2*, *PeNIP4-2*, *PeNIP1-1*, and *PeNIP1-2*. Additionally, the expression of some genes was suppressed, such as *PePIP2-8*, *PeTIP1-1*, *PeTIP4-1*, and *PeSIP2-1*. The expression patterns of some genes (*PeTIP1-3*, *PeNIP4-1*, *PeSIP1-1*, *PeSIP2-2*, and *PePIP2-7*) were not obvious. We verified and analyzed the expression of AQPs under drought stress. The results showed that most of the genes were induced by drought (Figure 10). Some AQP genes were suppressed under drought stress, such as *PeNIP4-1*, *PeNIP4-3*, *PeNIP5-1*, *PePIP2-2*, *PePIP2-7*, *PePIP2-8*, *PePIP2-11*, *PePIP2-12*, *PeSIP2-1*, *PeTIP1-1*, *PeTIP1-3*, *PeTIP2-1*, and *PeTIP4-1*. This gene expression trend is basically consistent with the results of the transcriptome. The result showed that most of the PeAQPs were able to respond to various abiotic stresses.

In the early stage, we performed transcript sequencing of three different fruit ripening stages in passion fruit [32], on the basis of which we analyzed the expression levels of the AQP family members (Figure 11). On the basis of the results of the heat map, most of the genes exhibited their highest expression levels during the first period, and their expression levels gradually decreased with decreasing fruit moisture during the next two periods. There was also a small number of family members whose expression level increased as the fruit matured, such as *PePIP1-2*, *PePIP2-7*, and *PeTIP2-2*. We also performed qRT-PCR verification, and the results showed that this gene expression trend was consistent with the transcriptome result (Figure 12). As the passion fruit ripened, its fruit moisture also decreased accordingly. This result indicated that the expression of AQPs decreased with increasing fruit maturity, which may be related to the water content in passion fruit.

Finally, we also analyzed the expression of AQPs in the leaves, roots, stems, and fruits of passion fruit (Figure 13). Among 27 members, 11 of them were mainly expressed in the leaves (*PeNIP1-1*, *PeNIP1-2*, *PeNIP4-2*, *PeNIP4-3*, *PePIP2-7*, *PePIP2-9*, *PePIP2-10*, *PePIP2-11*, *PeSIP2-1*, *PeTIP1-2*, and *PeNIP4-3*). Others were mainly concentrated in the leaves, roots, and stems, but to a lesser degree in the fruits, such as *PePIP1-2*, *PePIP2-7,* and *PeTIP2-2*. The results showed that each gene was expressed in different parts.

### 2.10. Overexpression of PeTIP3-2 Enhances Transgenic Arabidopsis Tolerance to Drought Stress

At present, the function of AQPs in plants has been discussed with respect to abiotic stress. In this study, we focused on *PeTIP3-2*, which is induced under drought stress (Figure 14). The transgenic plants were grown in 1/2MS and mannitol-added medium (simulating drought stress) for 14 days. The results of GUS staining showed that the transgenic plants were darker under drought stress, indicating that the expression of *PeTIP3-2* was induced under drought stress. We obtained three independent *PeTIP3-2* transgenic Arabidopsis lines, named L1, L2, and L3. The transgenic and wild-type plants (WT) grown for 4 weeks were treated without water for 8 and 11 days. The results showed that under drought stress, the survival rate of the transgenic plants was higher than that of the wild-type plants.

After 5 and 15 days of rehydration, most of the transgenic plants grew stronger than the wild-type plants. Physiological index measurements were performed on plants after being subjected to drought stress for 11 days and rewatering for 15 days. Under normal growth conditions, the MDA, ion leakage rate, PRO, and soluble sugar content of WT and transgenic plants were not significantly different. After drought stress and rewatering treatment, the MDA content and ion leakage rate in transgenic plants were lower, whereas PRO and soluble sugars were higher than those of WT. The results indicated that *PeTIP3-2* can improve the drought resistance of plants.

### 2.11. Osmotic Stress-Induced Activity Analysis in Transgenic Tobacco

We further transiently transformed tobacco with *PeTIP3-2* (Figure 15). The control and infected leaves were soaked in 1/2MS and 1/2MS + 18% PEG solutions, respectively. The samples were taken at 0, 2, 4, 8, 12, 24, 36, and 48 h, and then subjected to GUS staining. The results showed that the GUS staining of the other treatment groups was deepened after PEG treatment except for the 0 h group, indicating that *PeTIP3-2* could be induced by drought stress. Among them, the GUS staining of leaves was the darkest under 8, 12, and 24 h of treatment. This indicated that the gene was highly induced during these periods.

## 3. Discussion

Because of its short growth cycle and unique flavor, passion fruit is becoming increasingly popular worldwide. Biotic and abiotic stresses affect its growth and development processes. Therefore, mining the genes and analyzing its mechanism of resistance is of great significance for improving the resistance of the passion fruit. AQPs play important roles in plant growth and developmental processes [16]. In the current research, we conducted a genome-wide identification of PeAQPs, focusing on abiotic stress and fruit maturity stage, which will provide new clues for further research on the function of AQP genes and the genetic improvement in the passion fruit.

### 3.1. The Evolutionary Characteristics of the AQP Family in Plants

In this research, the classification of AQPs in passion fruit, like most higher plants, such as Arabidopsis, maize, rice [27,28,33] and barley [34], is performed through a division into four subfamilies. In addition, the members of the AQP family in cucumber, soybean, sesame, and tomato can be divided into five subfamilies: PIP, NIP, SIP, TIP, and XIP [16].

According to several reports, plant AQPs have different biochemical characteristics related to their functions [35]. The results showed that the physical and chemical properties of different subgroup members are different to some extent. The difference is mainly due to the presence of basic residues in the C-terminal domain of AQP proteins, as also observed in sweet orange AQPs [36]. Related reports have shown that the subcellular localization of NIPs and PIPs in most plants usually takes place on the plasma membrane, whereas TIPs and SIPs are usually located on the tonoplast and endoplasmic reticulum [37,38]. In this research, the subcellular localization results of PeAQP showed that only *PeNIP6-1* was predicted to be localized in the nucleus or mitochondria, whereas other members were on the cell membrane. This is consistent with the function of aquaporin in plants, which is mainly to transport water and other small molecules such as glycerol, CO_2_, and boron through biological membranes [4]. Among the TIP family members of melon, *CmTIP3;1* and *CmTIP5;1* are located in the mitochondria and chloroplasts, respectively [39]. In some plants, the aquaporin gene is located in two different regions, such as the *ZmPIP1;2* genes in maize, which are located in the plasma membrane and endoplasmic reticulum (ER) [40]. In tobacco, the *NtAQP1* gene has been reported to be located in the plasma membrane and the inner chloroplast membrane [41]. *AtTIP3;1* is located in both the plasma membrane and the vacuolar membrane [42]. An increasing number of studies have shown that the subcellular localization of plant genes is also closely related to abiotic stress factors [43].

The exon–intron structure is an important feature of genes. In this research, most family members contained 2–4 exons. The same pattern also exists in chickpeas. The research of Amit A. Deokar showed that most of the PIP, TIP, and NIP subfamily in chickpeas have 3, 2, and 4 introns, respectively [44]. Similar exon–intron patterns of the AQP family among species have also been reported in tomatoes, oranges, and soybeans [11,36,45]. In general, the intron-exon pattern is related to the conservation of gene family classification.

We also analyzed the chromosomal location and distribution of the AQP genes. There are nine chromosomes in the passion fruit. The 27 PeAQPs were located on seven chromosomes, which were chromosomes 1, 2, 3, 5, 6, 8, and 9. In chickpeas, *CaNIP1-6*, *CaNIP1-7*, and *CaNIP1-8* were simultaneously located on chromosome 1; *CaTIP4-2* and *CaTIP4-1* were located on chromosome 4; and *CaNIP1-3*, *CaNIP1-4*, and *CaNIP1-5* were all located on chromosome 6 [46].

### 3.2. The Function of AQP Promoters in Plants

Gene expression is regulated by promoters, which contain multiple cis-acting elements [47]. Promoters play an important role in regulating gene expression [48]. In this study, we selected the 2000 bp region upstream of the PeAQP CDS sequence and used PlantCARE to perform predictive analysis of the elements [49]. Most of the elements were light-responsive elements. It was found that light is one of the most important environmental factors regulating plant growth and development. Light regulates gene expression networks at many levels [50]. At the same time, there are related reports showing that light has an effect on the expression and function of the AQP family [51]. In addition to light-responsive elements, cis-acting elements related to abiotic stress were also identified, such as drought-responsive, cold-responsive, and salicylic acid-responsive elements. Secondly, there are some elements related to plant hormones such as the abscisic acid responsiveness element (ABRE), the MeJA-responsive element, gibberellin-responsive motifs, and the auxin-responsive element. This suggests that PeAQPs are regulated by stress responses and associated with plant growth and development.

### 3.3. The Functional Diversity of AQPs in Plants

In this study, we performed transcriptome sequencing analysis of four different stress treatments: drought, high salt, cold, and high temperature. Among the members of the passion fruit AQP family, most members can respond to different abiotic stresses. For example, *PePIP1-2* can respond to drought, high salt, and cold stress. *PeTIP2-2* and *PeTIP3-2* can be induced by and expressed in response to drought, high salt, cold, and high temperature. We performed qPCR detection and analysis of AQP gene expression in passion fruit under drought stress. Some members could be induced to express by drought stress, and one of the most highly induced genes, *PeTIP3-2*, was selected for the target gene with further functional verification. The results showed that *PeTIP3-2* could be induced in transgenic plants under drought stress. Most reports indicate that AQPs in plants can respond to abiotic stress; the banana AQP gene *MaPIP1;1* in transgenic Arabidopsis and banana can respond to drought and cold stresses [13,22]. *MaPIP2-7* and *MaSIP2-1* can respectively improve the drought resistance, salt, and cold tolerance in the transgenic bananas [4]. *MaTIP1;2* in banana can improve the drought and salt tolerance in transgenic Arabidopsis [52]. *CaTIP1-4*, *CaNIP1-4,* and *CaNIP1-8* in chickpeas can respond to high salt stress. *CaTIP1-1* and *CaPIP1-1* can improve the drought resistance and salt and high temperature tolerance of plants [44]. *TaAQP7* and *TaAQP8* of wheat can improve the drought and salt tolerance of transgenic tobacco [12,53]. Banana *MusaPIP2;6* can be highly induced by salt stress [54].

In addition, we also analyzed the expression of AQP genes during three fruit ripening periods [32], and the results showed that most of the AQP genes exhibited their highest expression levels in the first stage. As the fruit matures and the degree of water loss in the fruit increases, its expression gradually decreases, which may be consistent with the function of AQPs in transporting water. Some studies have shown that AQP is related to plant development. In grapes, some PIP genes have their maximum expression levels 60 to 80 days after flowering, among which, the expression levels of *VvPIP1-3* and *VvPIP2-1* decreased after 80 days of flowering, which was related to the increase in water resistance of grape xylem [55]. For *SlPIP1;2*, *SlPIP2;4*, *SlTIP1;1,* and *SlTIP2;1* in tomato, the expression level in the fruit remains high from 3 to 14 days after pollination [11]. *FaPIP1* and *FaPIP2* in strawberry are more highly expressed in the harder tissues, indicating that they may play a role in fruit firmness [9]. By detecting the tissue-specific and development-specific expression of tomato aquaporin genes, the contribution of aquaporin to water transport in leaves and fruit development is revealed [11].

Furthermore, the results of PeAQP expression in different plant tissues indicated that the members are expressed in each of the different plant tissues, and categorized into stems, leaves, roots, and fruits. The location of AQPs in different plants may be closely related to their function. For example, *MaPIP1;1* in banana is mainly expressed in the roots. It has been found that the roots of *MaPIP1;1* in transgenic Arabidopsis are longer than those in wild-type plants. The aquaporin gene *CmPIP2;10* in melon has the highest expression in the roots, and it has been found that it can transport solutes such as urea. Additionally, *CmNIP5;1* and *CmNIP5;2* are most highly expressed in roots and leaves [39]. The PIP and TIP family members in wheat are mostly expressed in the leaves [17]. It has been found in chickpeas that AQPs have the highest expression levels in meristems with high water flux, such as leaves, shoots, and roots [44]. This corresponds to the main function of plant AQPs, which is to transport water. Water-tolerant and -sensitive wheat were studied with and without drought stress treatment, respectively. The expression of different AQP genes in the tolerant wheat roots decreased under drought stress, while expression was induced in the sensitive plants. This shows that AQPs play an important role in transporting water in the root system of wheat [17]. The results also showed that the PIP subfamily in cotton has the highest expression in the roots, leaves, and fibers, indicating that it is involved in water transport and CO_2_ diffusion in cotton [16].

## 4. Materials and Methods

### 4.1. Identification of AQP Genes in Passion Fruit

The passion fruit genome sequences were retrieved from Phytozome V12.1 (https://ngdc.cncb.ac.cn/search/?dbId=gwh&q=GWHAZTM00000000) (accessed on 22 November 2021). Meanwhile, the AQP protein domain (PF00230) was downloaded from the PFAM database (http://pfam.xfam.org/) (accessed on 3 December 2021) and was applied to identify the candidate passion fruit AQP members. Additionally, blast analyses with all AQPs from the top three with the highest comparison values were used to further identify possible AQPs in the passion fruit database. Furthermore, the protein sequences identified by both of the above methods were integrated and parsed by manual editing to remove redundancies. The Arabidopsis and rice AQP protein sequences were retrieved and downloaded from the ensemble plant database (https://plants.ensembl.org/index.html) (accessed on 18 January 2022). The full-length protein sequences of the AQPs in passion fruit, Arabidopsis, and rice were aligned using ClustalX2 software (https://clustalx.software.informer.com/2.1/) (accessed on 22 January 2022). After the ALN file was generated, MEGA6 software (https://www.megasoftware.net/) (accessed on 19 January 2022) was used to build the neighbor-joining (NJ) phylogenetic tree [56]. The criteria were adopted with the pairwise deletion option and the Poisson correction model. A bootstrap test was performed with 1000 replicates. Finally, the PeAQP protein motifs were analyzed with the MEME tool (https://meme-suite.org/meme/tools/meme) (accessed on 19 January 2022) to compare whether the above PeAQPs had a common structural domain, and the exception members were removed. The members of the PeAQPs were finally considered by the screening of the above several methods.

### 4.2. Gene Identification, Gene Structure, and Chromosomal Locations

The protein iso-electric point (PI), molecular weight (MW), and the molecular formula for all candidate family members were computed using the ProtParam website (http://www.expasy.org/tools/protparam.html) (https://meme-suite.org/meme/tools/meme) (accessed on 11 February 2022). The NetPhos 3.1 Server (http://www.cbs.dtu.dk/services/NetPhos/) (accessed on 11 February 2022) was used to predict the protein phosphate site. Their subcellular localization was predicted using the WoLF PSORT software (https://www.genscript.com/wolf-psort.html) (accessed on 15 February 2022). The gene structure map was obtained using TBtools (Guangzhou, Guangdong, China) [57]. The combined visualization of the phylogenic tree, the motifs, and the gene structures of the PeAQPs were also obtained using TBtools, as were the visualizations of the chromosome localization and syntenic relationships [57].

### 4.3. Analysis of cis-Acting Elements of PeAQP Genes

A total of 2000 bp of genomic DNA sequence upstream of the transcriptional start site in each PeAQP was obtained from the PlantCARE database (http://bioinformatics.psb.ugent.be/webtools/plantcare/html/) (accessed on 18 February 2022), which was used to identify the cis-acting elements in the promoter region of PeAQPs [49].

### 4.4. Homology Modeling of PeAQP 3D Structure

The Swiss-Model interactive tool (https://swissmodel.expasy.org/interactive/) (accessed on 20 February 2022) was used to predict the 3D structure of the PeAQP proteins [58,59]. In addition, the PROCHECK test was used to check the 3D structure of the AQP proteins in the SAVES server (http://nihserver.mbi.ucla.edu/SAVES/) (accessed on 20 February 2022) and their 3D structure was displayed using Pymol software (https://pymol.org/2/) (accessed on 20 February 2022).

### 4.5. Analysis of Gene Protein Interaction Regulatory Network

First, the orthovenn2 tool was used (https://orthovenn2.bioinfotoolkits.net/home) (accessed on 22 February 2022) to identify the orthologous pairs between PeAQPs and AtAQPs. Second, the interaction networks in which PeAQPs were involved were identified based on the orthologous genes between the passion fruit and Arabidopsis using the AraNetV2 (http://www.inetbio.org/aranet/) (accessed on 3 March 2022). The STRING (http://string-db.org/cgi) (accessed on 5 March 2022) database and the predicted interaction network were displayed using Cytoscape software (https://cytoscape.org/) (accessed on 5 March 2022).

### 4.6. Gene Collinearity Analysis

The Multicollinearity Scanning Toolkit (MCScanX) (Fuzhou, Fujian, China) was used to analyze gene duplication events between species and within the same species [57].

### 4.7. Plant Materials and Growth Conditions

Healthy and virus-free passion fruit seedlings of purple fruit varieties were chosen. They were grown in soil under a growth chamber (30 °C; 200 μmol·m^−2^·s^−1^ light intensity; 12 h light/12 h dark cycle; 70% relative humidity) to a height of about 1 m and with 8–10 functional leaves, which were subjected to various abiotic stress treatments. For drought stress analysis, water was withheld from the treatment group to obtain soil moisture of 50% and 10%. For salt stress tolerance analysis, the seedlings were treated with 300 mM Nacl solution after 3 and 10 days. For the cold treatment, the plants were treated at 0 °C for 20 h and 48 h, respectively. For the high temperature treatment, the plants were treated at 45 °C for 2 h, 4 h, and 24 h.

### 4.8. RNA Extraction, Transcriptome Sequencing, and qRT-PCR

We froze the samples, and the total RNA was extracted from samples with different abiotic stress treatments and fruit maturity stages, and from the roots, stems, and leaves of the passion fruit, using the plant RNA isolation kit (Fuji, China, Chengdu) with three biological replicates. The cDNA was used for transcriptome sequencing analysis and qRT-PCR. Transcriptome data under different abiotic stress conditions and at different fruit maturity stages were used in this study. Primer sequences were designed using the Primer 5.0 tool. The expression of PeAQPs was detected by quantitative real-time polymerase chain reaction (qRT-PCR) analysis using SYBR^®^ Premix Ex Taq™ (TaKaRa, Japan, Tokyo) chemistry on Lightcycle-480 (Roche). Relative expression levels were calculated using the 2^−ΔΔCt^ method and normalized to the PeAQPs.

### 4.9. Heat Map

The genome-wide microarray data of PeAQPs in response to drought, salt, cold, and high-temperature stress and at various fruit maturity stages are shown in Appendix A. The transcript data of PeAQPs were analyzed using TBtools software (https://bio.tools/tbtools) (accessed on 22 January 2021). The normalized expression data were used to generate a heatmap using TBtools software [57].

### 4.10. Cloning and Vector Construction of PeTIP3-2

The full-length cDNA of *PeTIP3-2* was amplified from the passion fruit purple fruit varieties by reverse transcription–polymerase chain reaction (RT-PCR) based on the sequence information in the passion fruit genome database (https://bigd.big.ac.cn/gwh/) (accessed on 1 January 2021). Healthy passion fruit seedlings were used to construct a single-stranded cDNA template. PCR products were cloned into the pMD19-T vector (Promega, Madison, WI, USA) and sequenced on an ABI PRISM310 Genetic Analyzer (PerkinElmer Applied Biosystems, Foster City, CA, USA). The full-length cDNA of *PeTIP3-2* was assessed by DNAMAN software (https://www.lynnon.com/dnamanxl.html) (accessed on 22 January 2022) and BLAST (http://ncbi.nlm.nih.gov/blast) (accessed on 22 January 2021). The *PeTIP3-2* ORF, including engineered NcoI/SpeI restriction sites, was obtained using gene-specific primers. The PCR products were inserted into the pCAMBIA1304 expression vector to generate a *PeTIP3-2*-GFP fusion protein under the control of the CMV35S promoter.

### 4.11. Plant Transformation and Generation of Transgenic Plants

The pCAMBIA1304-*PeTIP3-2* construct was transferred into the Agrobacterium strain EHA105. Transgenic Arabidopsis was generated using the floral DIP-mediated infiltration method [60]. Seeds from T0 transgenic plants were selected on 1/2MS medium containing 25 mg/L of hygromycin B. Homozygous T3 lines were used for further functional investigation.

### 4.12. Drought and PEG Stress Treatment in WT and Transgenic Plants

The plants were subjected to drought stress tolerance analysis experiments. The wild-type and transgenic plants were grown at 23 °C for 4 weeks; water was withheld for 8 days and 11 days, and then they were given 5 days and 15 days of recovery, respectively, before being observed and photographed.

The 60-day-old tobacco leaves were used for transient expression experiments and treated with the detached leaves. Leaf discs with a diameter of 0.5 cm were cut out and floated in 1/2 MS liquid medium supplemented with 18% (*w*/*v*) PEG 6000 (osmotic stress) at 25 °C for 0, 2, 4, 8, 12, 24, 36, and 48 h, respectively. The leaf discs floating in 1/2 MS liquid medium were considered the control.

### 4.13. GUS Activity Detection

The 15-day transgenic Arabidopsis and transiently transformed tobacco in normal and mannitol-containing medium were stained with GUS and then photographed. Fresh samples (small plants) were placed in X-Gluc solution (GBT, St. Louis, MO, USA) [61] for histochemical analysis. The seedlings were placed in GUS staining solution containing 50 mM sodium phosphate (pH 7.0), 0.5 mM potassium ferricyanide, 0.5 mM potassium ferrocyanide, 10 mM EDTA, 0.1% Triton X-100, and 1 mM X-Gluc, then incubated at 37 °C for 24 h. Fluorometric determination of GUS enzyme activity was conducted using 4-methylumbelliferyl glucuronide.

## 5. Conclusions

Aquaporins in plants can transport water and small molecules, and are widely reported to play a role in the abiotic stress resistance of plants. We identified 27 aquaporin members from the genome of the passion fruit. Analyses were performed with respect to evolutionary tree, structural domains, promoter cis-acting elements, and inter-species, and intra-species collinearity. PeAQPs are able to respond to abiotic stresses such as drought, high salinity, cold, and high temperature. This is expressed in different organizations of the passion fruit. As the fruit matures, the expression of PeAQPs gradually decreases, indicating that it is related to water transport. We focused on the gene *PeTIP3-2*, which is induced by drought stress. Under drought stress, *PeTIP3-2* transgenic Arabidopsis exhibited significantly deeper GUS staining than the control. The survival rate and resistance of transgenic Arabidopsis were higher than those of the wild type. These results indicate that *PeTIP3-2* may be used as a candidate gene for the development of drought-tolerant crops. Our research can provide a basis for further understanding the function of PeAQPs and analyzing their mechanism of action in improving plant stress resistance.

## Figures and Tables

**Figure 1 ijms-23-05720-f001:**
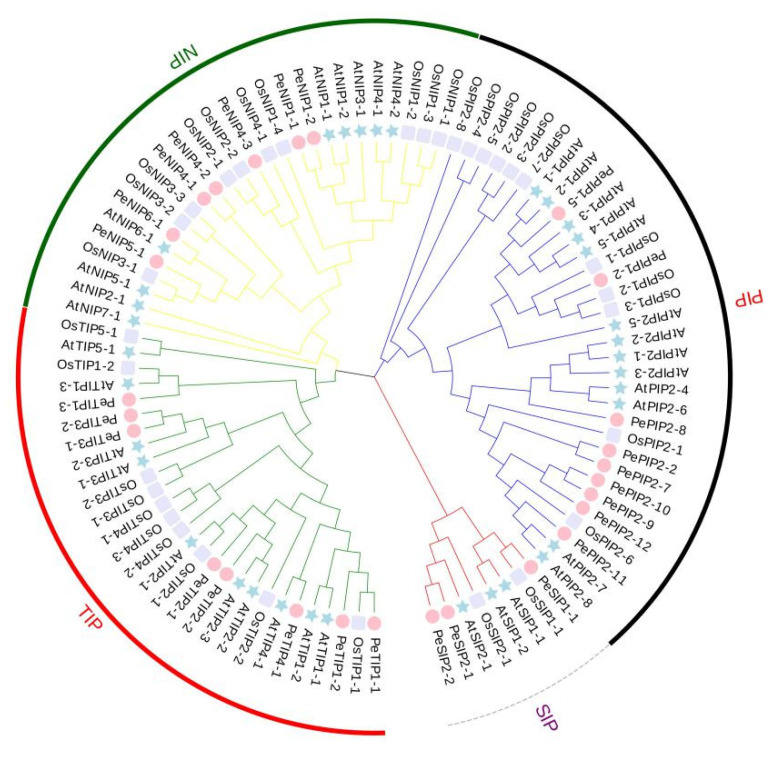
Phylogenetic relationships of AQPs among the passion fruit, Arabidopsis, and rice. The tree was constructed by ClustalX and MEGA 7.0 using the neighbor-joining method with 1000 bootstraps. The four sub-families (TIPs, NIPs, SIPs, PIPs) were classified and displayed in different groups by different colored outer rings. The circle represents the AQPs in passion fruit; the five-pointed star represents the AQPs in Arabidopsis; the square represents the AQPs in rice.

**Figure 2 ijms-23-05720-f002:**
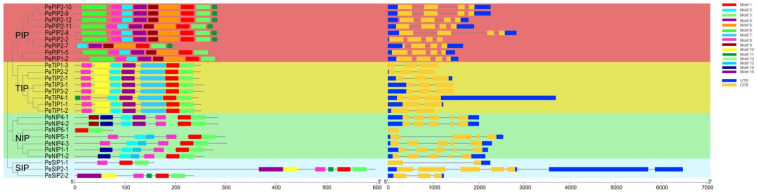
Phylogenetic clustering, conserved motifs, and exon–intron organization of *PeAQPs***.** The phylogenetic tree was constructed based on *PeAQPs* and divided into four different color groups. A total of 15 motifs were denoted by different colored boxes. Untranslated (UTR) 5′ and 3′ regions are indicated by blue boxes, exon regions are indicated by yellow frames, and intron regions are indicated by black lines.

**Figure 3 ijms-23-05720-f003:**
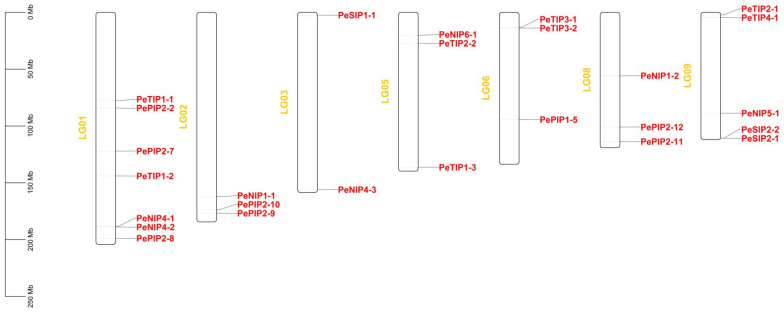
Distribution of the identified 27 *PeAQPs* across the passion fruit genome. The figure shows that *PeAQPs* are located on No. 1, 2, 3, 5, 6, 8, 9 chromosomes.

**Figure 4 ijms-23-05720-f004:**
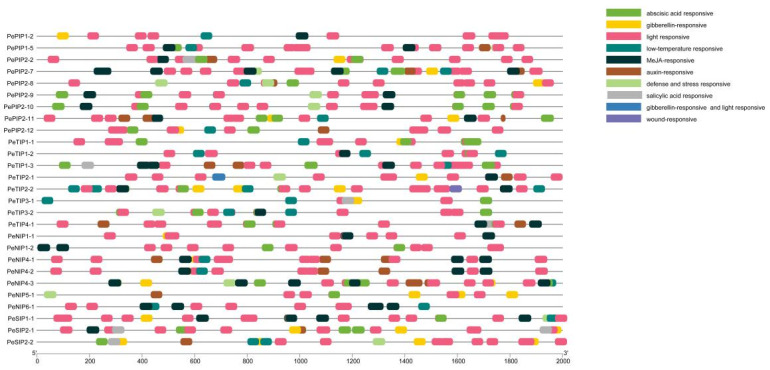
The cis-acting element in the 2000 bp promoter upstream of *PeAQP* gene. Known cis-acting elements are shown in the right.

**Figure 5 ijms-23-05720-f005:**
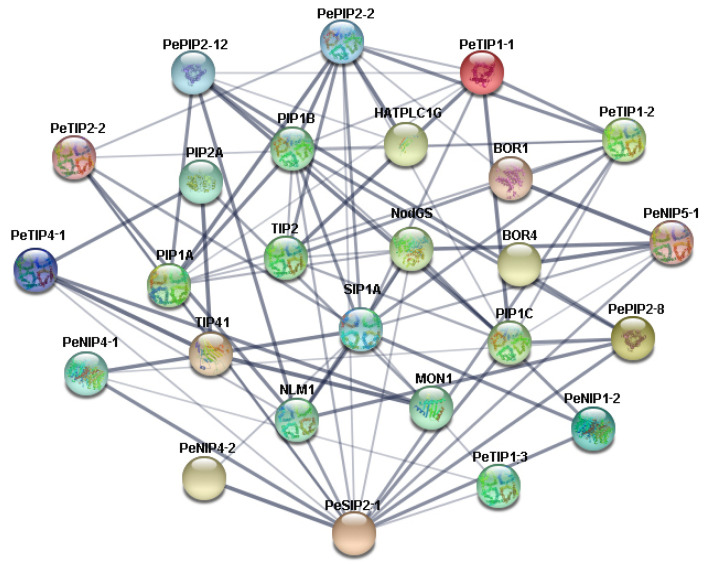
Predicted protein–protein interaction networks of PeAQP proteins with other passion fruit proteins using the STRING tool. The circles on the outside represent PeAQP proteins, and the other circles shows proteins that interact with PeAQPs. The gray connections between the circles represent the interactions between proteins.

**Figure 6 ijms-23-05720-f006:**
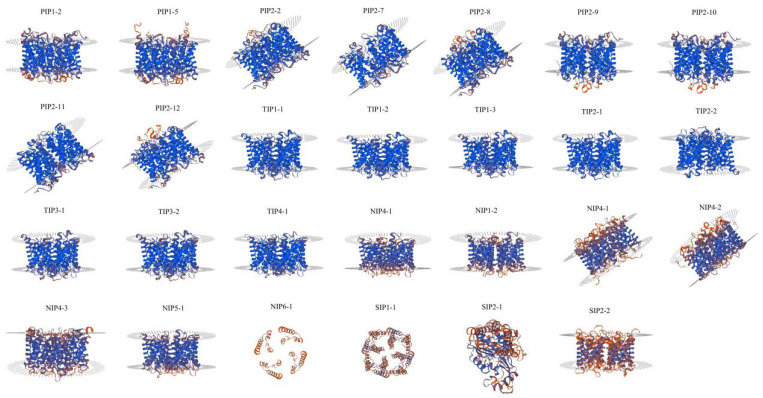
The 3D structure modeling of PeAQP proteins.

**Figure 7 ijms-23-05720-f007:**
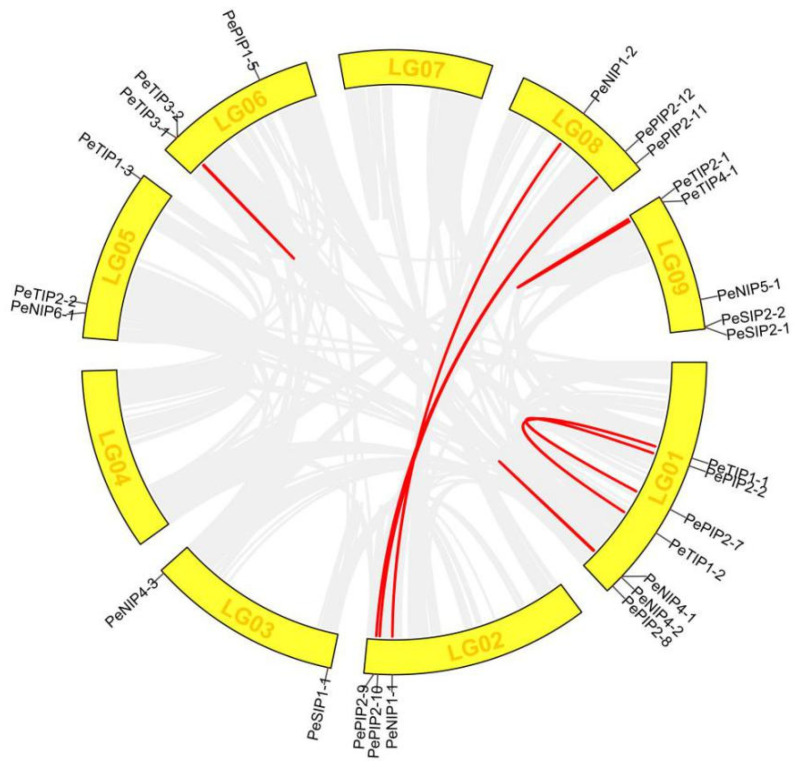
The synteny analysis of AQPs in the genomes of passion fruit. The red line represents the repeated *PeAQP* gene pair.

**Figure 8 ijms-23-05720-f008:**
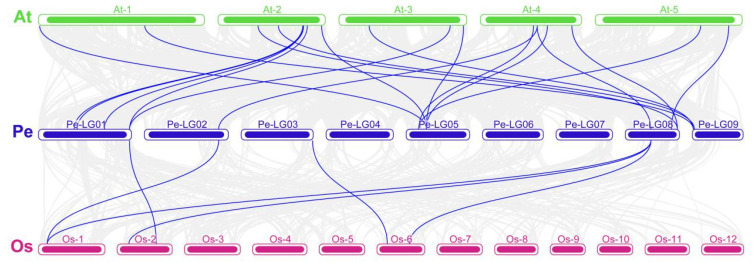
The synteny analysis of AQPs in the genomes between the passion fruit, rice, and Arabidopsis. The blue lines represent homologous gene pairs between the three species.

**Figure 9 ijms-23-05720-f009:**
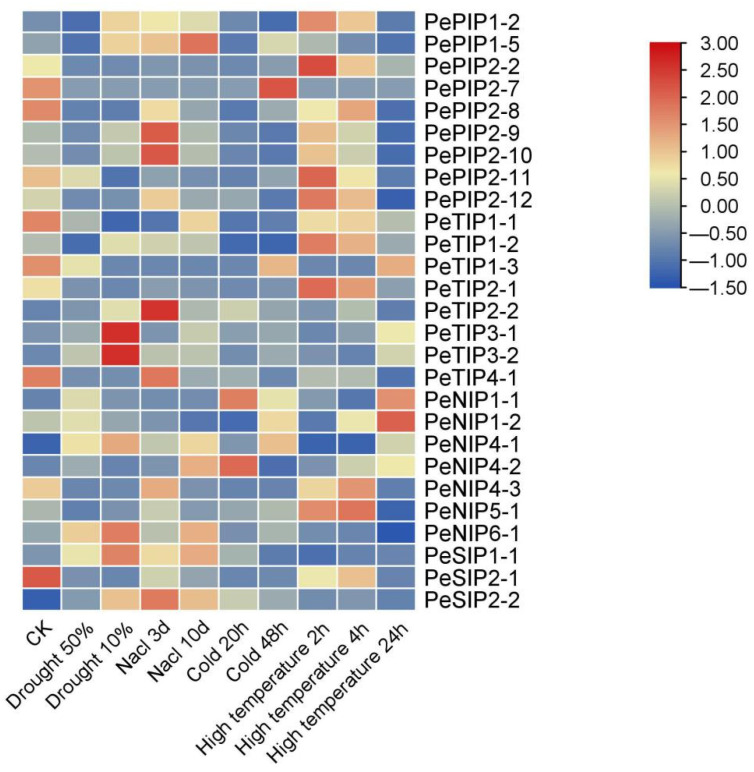
Expression profiles of PeAQPs genes responding to drought, salt, cold, and high temperature. The details are shown in Appendix A. Blue indicates a low expression level and red indicates a high expression level. The heat map was generated using TBtools.

**Figure 10 ijms-23-05720-f010:**
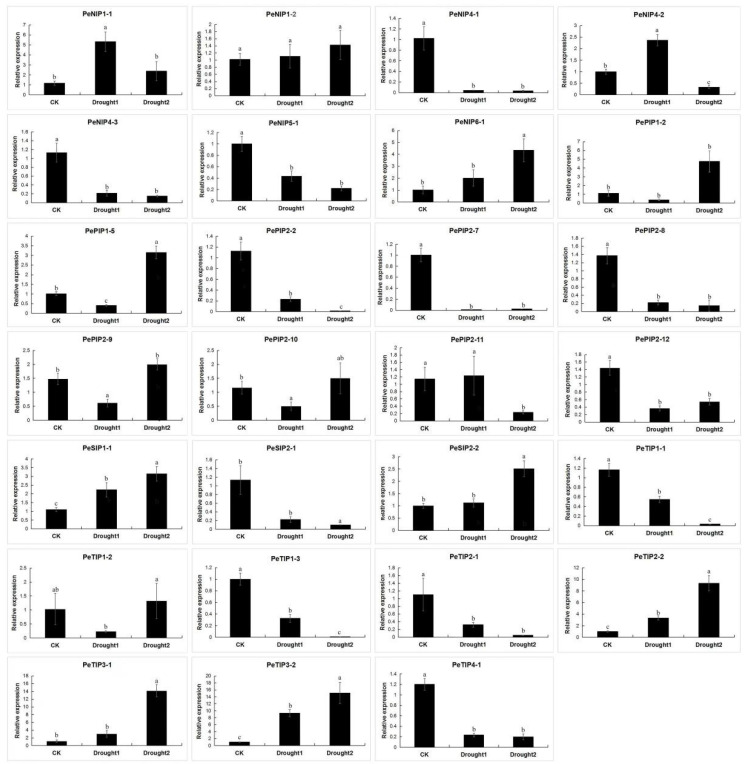
Expression analysis of 27 PeAQPs under drought stress in the passion fruit. The details are shown in Appendix A. Data are means ± SD of *n* = 3 biological replicates. Means denoted by the same letter are not significantly different at *p* < 0.05 as determined by Duncan’s multiple range test.

**Figure 11 ijms-23-05720-f011:**
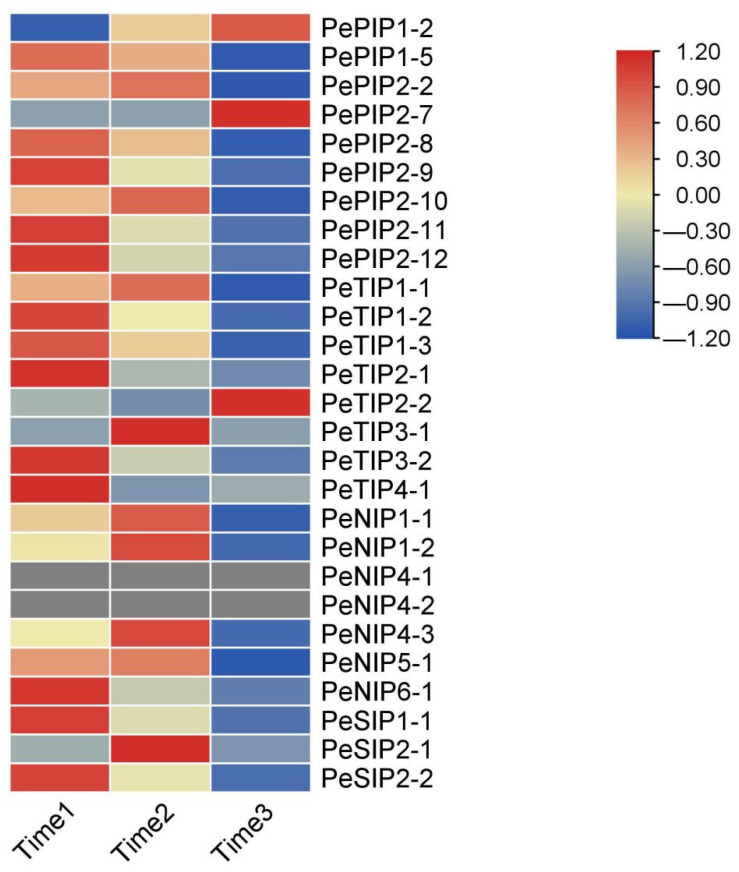
Expression profiles of PeAQPs genes during three fruit ripening periods. The details are shown in Appendix A. Blue indicates a low expression level and red indicates a high expression level. The heat map was generated using TBtools.

**Figure 12 ijms-23-05720-f012:**
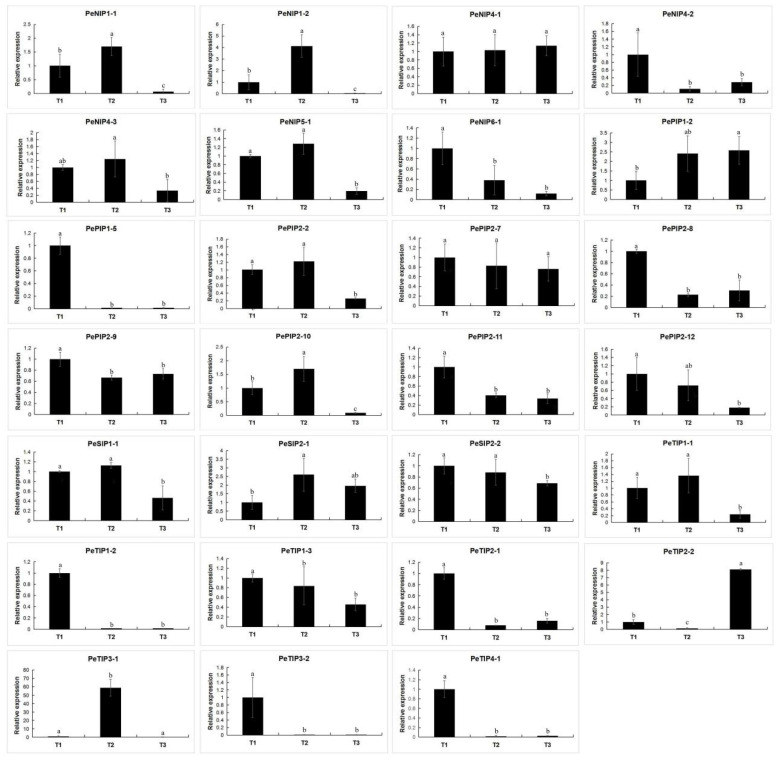
Expression analysis of 27 PeAQPs during three fruit ripening periods in the passion fruit. The details are shown in Appendix A. Data are means ± SD of *n* = 3 biological replicates. Means denoted by the same letter are not significantly different at *p* < 0.05 as determined by Duncan’s multiple range test.

**Figure 13 ijms-23-05720-f013:**
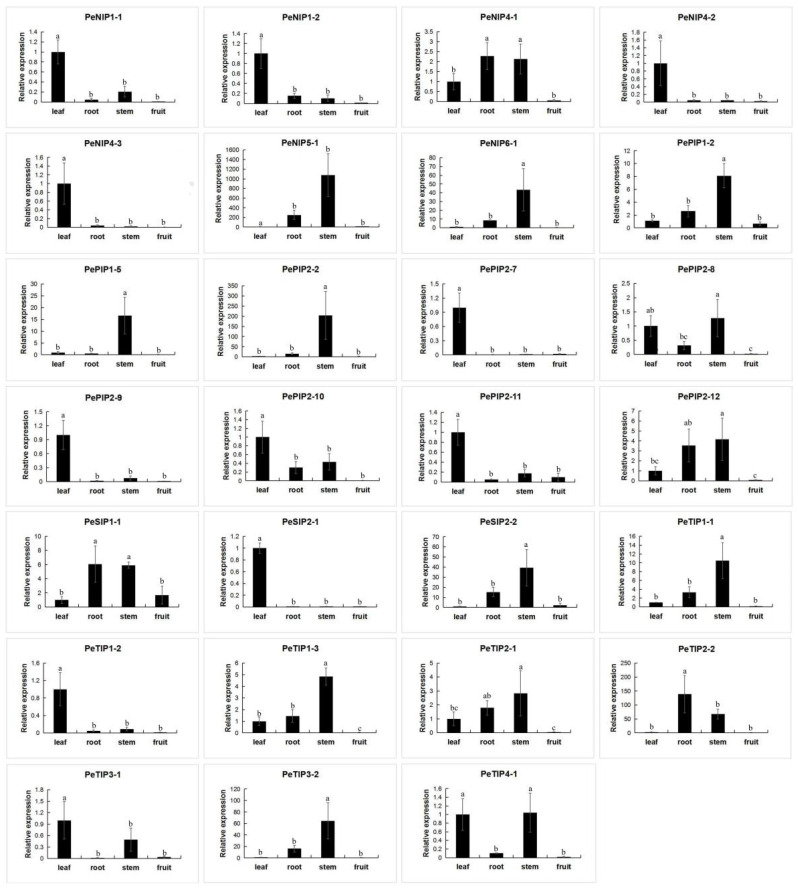
Expression analysis of 27 PeAQPs in the leaf, root, stem, and fruit of the passion fruit. The details are shown in Appendix A. Data are means ± SD of *n* = 3 biological replicates. Means denoted by the same letter are not significantly different at *p* < 0.05 as determined by Duncan’s multiple range test.

**Figure 14 ijms-23-05720-f014:**
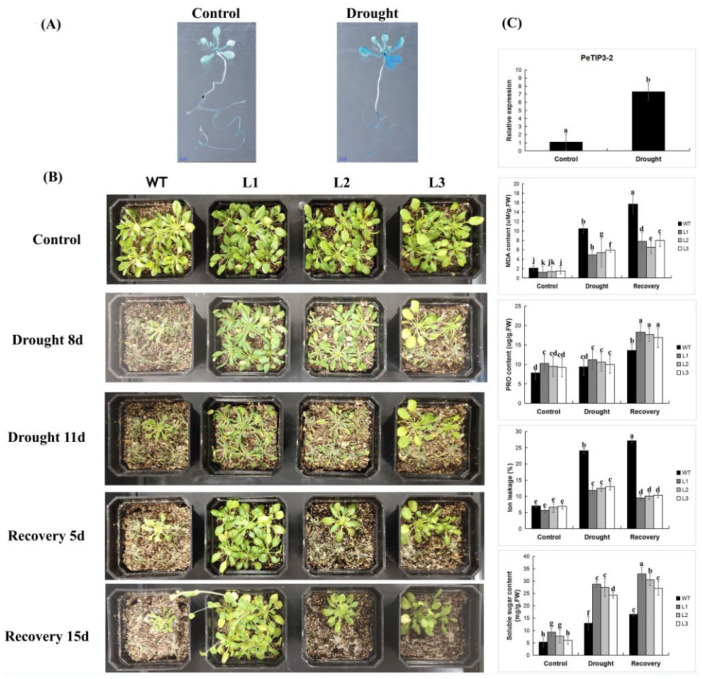
Phenotype differences and comparisons of physiological indices between WT and transgenic plants under normal, drought, and recovery conditions. (**A**) GUS histochemical staining of transgenic Arabidopsis under normal and drought stress conditions. (**B**) Phenotype between WT and transgenic plants under normal, drought, and recovery conditions. (**C**) The relative expression of *PeTIP3-2* under normal and drought conditions and comparisons of physiological indices between WT and transgenic plants under normal, drought, and recovery conditions. Means denoted by the same letter are not significantly different.

**Figure 15 ijms-23-05720-f015:**
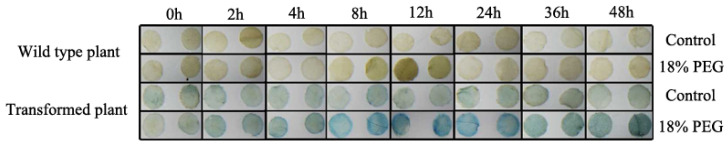
GUS staining of detached leaves of transgenic tobacco under normal and PEG treatment conditions. The leaf discs (diameter 0.5 cm) of WT and transgenic tobacco plants (3 replicates per treatment) were incubated in liquid 1/2 MS medium supplemented with 18% PEG6000 (w/v) for 2, 4, 8, 12, 24, 36, and 48 h; leaf discs floated in liquid 1/2 MS medium were used as control. The leaf discs were then incubated in staining solution at 37 °C for 24 h. Finally, the samples were observed and photographed after decolorization.

**Table 1 ijms-23-05720-t001:** Basic information of AQP genes identified in passion fruit.

Gene	Gene ID	CDS Length (bp)	Protein Length (aa)	Molecular Formula	MW (Da)	pI	Number of Phosphate Sites	Subcellular Localization
*PePIP1-2*	P_edulia060015940.g	864	287	C_1421_H_2155_N_359_O_380_S_11_	30,700.58	8.22	Ser:11 Thr:8 Tyr:5	Cell membrane
*PePIP1-5*	P_edulia020006715.g	858	285	C_1416_H_2163_N_357_O_377_S_9_	30,508.46	8.21	Ser:9 Thr:6 Tyr:5	Cell membrane
*PePIP2-2*	P_edulia010001259.g	771	256	C_1276_H_1975_N_329_O_332_S_5_	27,397.02	7.11	Ser:4 Thr:3 Tyr:3	Cell membrane
*PePIP2-7*	P_edulia020006513.g	597	198	C979H1497N257O255S6	21,139.58	9.71	Ser:10 Thr:4 Tyr:3	Cell membrane
*PePIP2-8*	P_edulia010005210.g	741	246	C_1203_H_1854_N_292_O_319_S_7_	25,736.14	5.57	Ser:7 Thr:5 Tyr:2	Cell membrane
*PePIP2-9*	P_edulia020006819.g	858	285	C_1416_H_2163_N_357_O_377_S_9_	30,508.46	8.21	Ser:9 Thr:6 Tyr:5	Cell membrane
*PePIP2-10*	P_edulia080019927.g	791	252	C_1201_H_1826_N_294_O_325_S_5_	25,743.78	5.55	Ser:5 Thr:5 Tyr:1	Cell membrane
*PePIP2-11*	P_edulia080019415.g	744	247	C_1177_H_1803_N_285_O_320_S_5_	25,226.28	5.86	Ser:9 Thr:4 Tyr:4	Cell membrane
*PePIP2-12*	P_edulia010001027.g	753	250	C_1175_H_1800_N_278_O_322_S_3_	25,069.07	4.93	Ser:12 Thr:1 Tyr:3	Cell membrane
*PeTIP1-1*	P_edulia010000969.g	758	252	C_1210_H_1834_N_296_O_326_S_5_	25,903.96	5.71	Ser:8 Thr:5 Tyr:1	Cell membrane
*PeTIP1-2*	P_edulia010001331.g	477	158	C_826_H_1244_N_196_O_201_S_6_	17,328.52	9.3	Ser:2 Thr:4 Tyr:2	Cell membrane
*PeTIP1-3*	P_edulia050012817.g	1794	597	C_2924_H_4612_N_802_O_845_S_29_	65,451.39	7.3	Ser:29 Thr:3 Tyr:0	Cell membrane
*PeTIP2-1*	P_edulia090020609.g	711	236	C_1195_H_1882_N_304_O_305_S_9_	25,676.48	10.09	Ser:11 Thr:5 Tyr:0	Cell membrane
*PeTIP2-2*	P_edulia050011915.g	900	299	C_1427_H_2243_N_371_O_399_S_10_	31,301.35	7.69	Ser:13 Thr:8 Tyr:3	Cell membrane
*PeTIP3-1*	P_edulia060014309.g	858	285	C_1384_H_2188_N_348_O_386_S_12_	30,263.42	6.43	Ser:13 Thr:6 Tyr:6	Cell membrane
*PeTIP3-2*	P_edulia060014363.g	831	276	C_1333_H_2091_N_339_O_381_S_12_	29,347.03	8.24	Ser:15 Thr:8 Tyr:3	Cell membrane
*PeTIP4-1*	P_edulia090020949.g	756	251	C_1195_H_1815_N_293_O_329_S_9_	25,838.86	4.97	Ser:9 Thr:4 Tyr:1	Cell membrane
*PeNIP1-1*	P_edulia080019109.g	861	286	C1417H2156N360O373S8	30,459.37	8.84	Ser:13 Thr:7 Tyr:2	Cell membrane
*PeNIP1-2*	P_edulia010004282.g	858	278	C_1370_H_2103_N_347_O_365_S_10_	29,595.47	8.61	Ser:11 Thr:5 Tyr:4	Cell membrane
*PeNIP4-1*	P_edulia010004322.g	837	285	C_1408_H_2164_N_358_O_379_S_9_	30,459.38	8.22	Ser:10 Thr:8 Tyr:4	Cell membrane
*PeNIP4-2*	P_edulia030009116.g	909	302	C_1444_H_2270_N_370_O_410_S_14_	31,822.98	7.77	Ser:22 Thr:9 Tyr:2	Cell membrane
*PeNIP4-3*	P_edulia090022062.g	234	77	C_351_H_569_N_103_O_107_S_4_	8072.24	9.69	Ser:3 Thr:7 Tyr:0	Nucleus
*PeNIP5-1*	P_edulia050011768.g	861	286	C_1421_H_2186_N_360_O_384_S_9_	30,745.71	7.67	Ser:6 Thr:10 Tyr:4	Cell membrane
*PeNIP6-1*	P_eduliaContig210022875.g	864	273	C_1365_H_2091_N_341_O_361_S_9_	29,343.23	8.99	Ser:8 Thr:10 Tyr:4	Cell membrane
*PeSIP1-1*	P_edulia030007640.g	858	285	C_1383_H_2188_N_346_O_386_S_12_	30,223.39	6.36	Ser:13 Thr:6 Tyr:6	Cell membrane
*PeSIP2-1*	P_edulia090022232.g	773	257	C_1259_H_1968_N_312_O_351_S_11_	27,444.02	6.82	Ser:17 Thr:6 Tyr:2	Cell membrane
*PeSIP2-2*	P_edulia090022235.g	771	256	C_1275_H_1973_N_329_O_332_S_5_	27,383	7.11	Ser:4 Thr:3 Tyr:2	Cell membrane

## Data Availability

Not applicable.

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
