# Peer review of "Genome-Wide Identification and Expression Analyses of the Aquaporin Gene Family in Passion Fruit (Passiflora edulis), Revealing PeTIP3-2 to Be Involved in Drought Stress"

_ijms, 2022, doi:10.3390/ijms23105720_

Round 1

Reviewer 1 Report

The present manuscript constitutes the first genome-wide identification of the P. edulis aquaporin protein family and it studies the role of the different identified isoforms under abiotic stresses. The study provides a huge and impressive amount of data that is well-appreciated. 

Generally, I consider that methodology and experimental design are correct. However, I can not accept the manuscript either in the present form or with minor revisions because the language and drafting do not meet the requirements for publishing. Below, I give some tips which I think that help in improving the manuscript, but the writing must be revised on the whole.

-Introduction:

Apart from numerous misprints, the introduction may be improved by adding more accurate references (i.e. line 48-59 references are not specific, line 78) and by improving the last paragraph with the justification of the study (i.e. In regulating plants? this is not correct).

-Results:

In 2.2. you described the homology between Passiflora and Arabidopsis AQPs, however, you missed comparative with rice AQPs. I understand that Passiflora is more related to Arabidopsis, but you need to describe all the results.

In 2.7 the last sentence may correspond to the discussion, not to results.

Figures 10, 12, and 13 do not have the correct size, letters are too small. You could evaluate the possibility of putting some more figures as supplemental material.

Discussion

Overall, connectors between sentences are missing, making it difficult to understand the general meaning of the paragraphs. It is very descriptive and in some cases, it seems like a new description of the results instead of a discussion.

Author Response

Response to Reviewer 1 Comments

Thank you very much for your valuable suggestions and pointing out the errors in the manuscript . We have corrected and supplemented the corresponding errors, and commissioned the MDPI agency to polish the language of the manuscript. Please check and help us to point out if there is any problem. We also upload the revised manuscripts and comments. Please see the attachment.

Point 1:Apart from numerous misprints, the introduction may be improved by adding more accurate references (i.e. line 48-59 references are not specific, line 78) and by improving the last paragraph with the justification of the study (i.e. In regulating plants? this is not correct).

Response 1:Thank you for your suggestion. We have revised the introduction and fixed some syntax errors. The corresponding modifications have been made in lines 63-66, 82, 84, 91, 92, 106, 107, 120-122, respectively. The last paragraph has been changed into “This provides useful information on the genetic improvement of passion fruit quality and resistance to abiotic stress, and lays a good foundation for the regulating mechanisms of PeAQPs.”

Point 2:In 2.2. you described the homology between Passiflora and Arabidopsis AQPs, however, you missed comparative with rice AQPs. I understand that Passiflora is more related to Arabidopsis, but you need to describe all the results.

Response 2:Thanks for your suggestion, we have added the passion fruit and rice evolution alignment results in lines 171 to 177 of the result.

Point 3:In 2.7 the last sentence may correspond to the discussion, not to results.

Response 3:We have removed the sentence in 2.7.

Point 4:Figures 10, 12, and 13 do not have the correct size, letters are too small. You could evaluate the possibility of putting some more figures as supplemental material.

Response 4:We have adjusted the image and will re-upload a new image.

Point 5:Overall, connectors between sentences are missing, making it difficult to understand the general meaning of the paragraphs. It is very descriptive and in some cases, it seems like a new description of the results instead of a discussion.

Response 5:Thank you very much for your suggestion, we have revised the discussion . For example, adding some sentences to improve the connection. Please check if the modification is appropriate.

Reviewer 2 Report

In the abstract, you have mentioned " In the three different fruit 
ripening stages, most of the AQPs expressed highest in the first stage:

Can you be more specific in writing the result, how many AQPs 

Highest( can you specify units, how many ) 

Also regarding the validation study in Arabidopsis 

The results indicated that PeTIP3
can improve the drought resistance of plants.

Can you add a line mentioning the values and traits which support your claim for drought tolerance?

In the introduction, you mention passiflora, while in the title its passion fruit,  is categorized based on various clours of passion fruit.

Please be uniform in mentioning the fruit or flower under study.

Introduction can be made more precise and to the point.

Overall the study has merit but the presentation lacks quality, so rewrite the text and take help from an English language expert  

Author Response

Response to Reviewer 2 Comments

Thank you very much for your valuable suggestions and pointing out the errors in the manuscript . We have corrected and supplemented the corresponding errors, and commissioned the MDPI agency to polish the language of the manuscript. Please check and help us to point out if there is any problem. We also upload the revised manuscripts and comments. Please see the attachment.

Point 1:In the abstract, you have mentioned " In the three different fruit ripening stages, most of the AQPs expressed highest in the first stage:Can you be more specific in writing the result, how many AQPs Highest( can you specify units, how many ) 

Response 1:We have revised this conclusion in line 35-36 of the abstract.

Point 2:Also regarding the validation study in Arabidopsis The results indicated that PeTIP3 can improve the drought resistance of plants.Can you add a line mentioning the values and traits which support your claim for drought tolerance?

Response 2:We have revised this conclusion in line 39-40 of the abstract.

Point 3:In the introduction, you mention passiflora, while in the title its passion fruit,  is categorized based on various clours of passion fruit.

Response 3:In the manuscript, Passiflora refers to the genus of the plant, while the scientific name is passionfruit, which has been revised throughout the text.

Point 4:Please be uniform in mentioning the fruit or flower under study.

Response 4:We have changed the name to passion fruit in the full text.

Point 5:Introduction can be made more precise and to the point.

Response 5:We have added some sentences to increase the connection of the language.

Point 6: Overall the study has merit but the presentation lacks quality, so rewrite the text and take help from an English language expert  

Response 6:Thanks for your suggestion. We have commissioned MDPI to revise the manuscript. Please check the full text and help us point out any problems.

Round 2

Reviewer 1 Report

The authors have clearly improved the manuscript with their corrections, and they have addressed all the suggestions I made in my first review. Thus, I consider that it can be published in the present form. As a suggestion, however, I have to say that the quality of the figures can still be ameliorated.